# What's Behind the Mask: Estimating Uncertainty in Image-to-Image Problems

## Abstract

Estimating uncertainty in image-to-image networks is an important task, particularly as such networks are being increasingly deployed in the biological and medical imaging realms. In this paper, we introduce a new approach to this problem based on masking. Given an existing image-to-image network, our approach computes a mask such that the distance between the masked reconstructed image and the masked true image is guaranteed to be less than a specified threshold, with high probability. The mask thus identifies the more certain regions of the reconstructed image. Our approach is agnostic to the underlying image-to-image network, and only requires triples of the input (degraded), reconstructed and true images for training. Furthermore, our method is agnostic to the distance metric used. As a result, one can use $L_p$-style distances or perceptual distances like LPIPS, which contrasts with interval-based approaches to uncertainty. Our theoretical guarantees derive from a conformal calibration procedure. We evaluate our mask-based approach to uncertainty on image colorization, image completion, and super-resolution tasks, demonstrating high quality performance on each.

## 1 Introduction

Deep Learning has been very successful in many applications, spanning computer vision, speech recognition, natural language processing, and beyond. For many years, researchers were mainly content to develop new techniques to achieve unprecedented accuracy, without concern for understanding the uncertainty implicit in such models. More recently, however, there has been a concerted effort within the research community to understand and quantify the uncertainty of deep models.

This paper addresses the problem of estimating uncertainty in the realm of image-to-image (sometimes referred to as image reconstruction) tasks. Such tasks include super-resolution, deblurring, colorization, and image completion, amongst others. Computing the uncertainty is important generally, but is particularly so in application domains such as biological and medical imaging, in which fidelity to the ground truth is paramount. If there is an area of the reconstructed image where such fidelity is unlikely or unreliable due to high uncertainty, this is crucial to convey.

Our approach to uncertainty estimation is based on masking. Specifically, we are interested in the possibility of computing a mask such that the uncertain regions in the image are masked out. More formally, we would like the distance between the masked reconstructed image and the masked true image to be small in expectation. Ideally, the method should be agnostic to the choice of distance function, which should be dictated by the application. A high level overview of our approach is illustrated in Figure 1.

We show a direct connection between the mask and a theoretically well-founded definition of uncertainty that matches image-to-image tasks. We then derive an algorithm for computing such a mask which can apply to any existing (i.e. pre-trained) image-to-image network, and any distance function between image pairs. All that is required for training the mask network is triplets of the input (degraded), reconstructed and true images. Using a procedure based on conformal prediction (Angelopoulos & Bates, 2021a), we can guarantee that the masks so produced satisfy the following criterion: the distance between the masked reconstructed image and the masked true image is guaranteed to be less than a specified threshold, with high probability. We demonstrate the power of the method on image colorization, image completion, and super-resolution tasks.

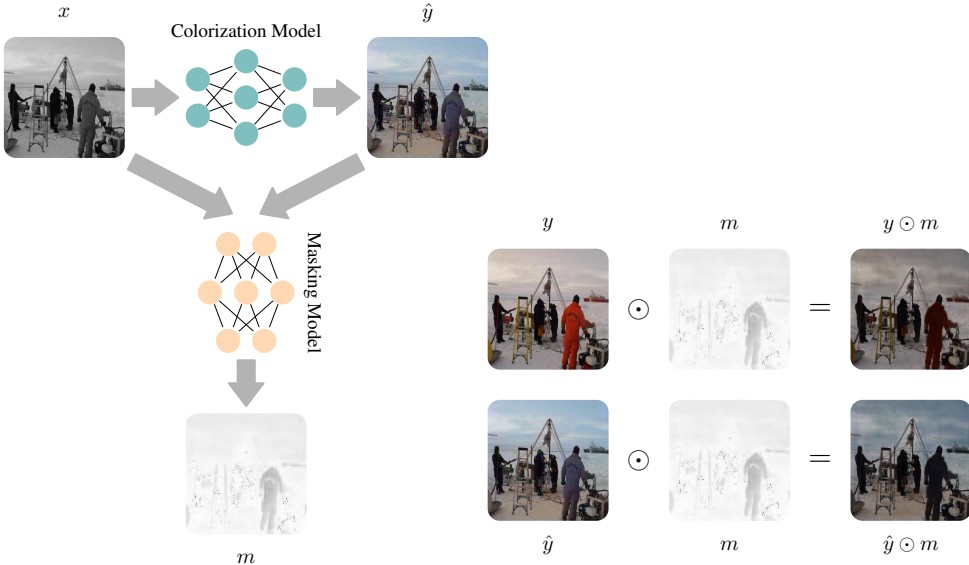

Figure 1: A high level view of our approach: $x$ is the degraded input (colorization in this example) to the regression model. $x$ and $\hat{y}$ (the colorized image) serve as the input to the masking model which predicts the uncertainty mask $m$. The ground-truth image is denoted by $y$. With high probability, the distortion between the masked images, $y \odot m$ and $\hat{y} \odot m$ is low.

Our contributions are as follows:

- We present an approach to uncertainty computation based on masking. We show that moving from binary to continuous masks preserves the connection with a theoretically well-founded definition of uncertainty that is relevant for image-to-image tasks.

- We derive an algorithm for computing the masks which works for arbitrary image-to-image networks, and any distance function between image pairs. The masks are guaranteed to yield a small distance between masked ground truth and reconstructed images, with high probability.

- We demonstrate the effectiveness of the method on image colorization, image completion, and super-resolution tasks attaining high quality performance on each.

NOTATIONS

Throughout the paper we represent an image as a column-stacked vector $x \in \mathbb{R}^n$. We use $x_k$ to denote the $k$th image of a collection, while $x_{(i)}$ marks the $i$th entry of $x$. We define $\vec{0} \in \mathbb{R}^n$ and $\vec{1} \in \mathbb{R}^n$ to be vectors of all zeros and ones respectively. The operation $x \odot y$ stands for the Hadamard (point-wise) product between $x$ and $y$. For $p \geq 1$, we denote the $\ell_p$-norm of $x$ by $\|x\|_p^p \triangleq \sum_{i=1}^n |x_{(i)}|^p$. The symbol $d(\cdot, \cdot)$ represents a general distortion measure between two images. We define a continuous *mask* as a vector $m \in [0, 1]^n$, where the average size of a mask equals $\|\vec{1} - m\|_1 / n$ such that the mask $\vec{1}$ (no-masking) has size 0 while the mask $\vec{0}$ has size of 1. Given a mask $m$, we define $d_m(x, y) \triangleq d(m \odot x, m \odot y)$ as the distortion measure between the masked versions of images $x$ and $y$. For a natural number $n \geq 1$ we define the set $[n] \triangleq \{1, ..., n\}$.

## 2 RELATED WORK

**Bayesian Uncertainty Quantification** The Bayesian paradigm defines uncertainty by assuming a distribution over the model parameters and/or activation functions. The most prevalent approach is Bayesian neural networks (MacKay, 1992; Valentin Jospin et al., 2020; Izmailov et al., 2020), which are stochastic models trained using Bayesian inference. Yet, as the number of model parameters has

grown rapidly, computing the exact posteriors has became computationally intractable. This short-coming has led to the development of approximation methods such as Monte Carlo dropout (Gal & Ghahramani, 2016; Gal et al., 2017a), stochastic gradient Markov chain Monte Carlo (Salimans et al., 2015; Chen et al., 2014), Laplacian approximations (Ritter et al., 2018) and variational inference (Blundell et al., 2015; Louizos & Welling, 2017; Posch et al., 2019). Alternative Bayesian techniques include deep Gaussian processes (Damianou & Lawrence, 2013), deep ensembles (Ashukha et al., 2020; Hu et al., 2019), and deep Bayesian active learning (Gal et al., 2017b), to name just a few. For a recent comprehensive review on Bayesian uncertainty quantification we refer the readers to Abdar et al. (2021).

**Distribution-Free Methods and Conformal Prediction**  Unlike Bayesian methods, the frequentist approach assumes the true model parameters are fixed with no underlying distribution. Examples of such distribution-free techniques are model ensembles (Lakshminarayanan et al., 2017; Pearce et al., 2018), bootstrap (Kim et al., 2020; Alaa & Van Der Schaar, 2020), interval regression (Pearce et al., 2018; Kivaranovic et al., 2020; Wu et al., 2021) and quantile regression (Gasthaus et al., 2019; Romano et al., 2019). An important distribution-free technique which is most relevant to our work is conformal prediction (Angelopoulos & Bates, 2021b; Shafer & Vovk, 2008). The conformal approach relies on a labeled calibration dataset to convert point estimations into prediction regions. Conformal methods can be used with any estimator, require no retraining, are computationally efficient and provide coverage guarantees in finite samples (Lei et al., 2018). Recent development includes conformalized quantile regression (Romano et al., 2019; Sesia & Candès, 2020; Angelopoulos et al., 2022b), conformal risk control (Angelopoulos et al., 2022a; Bates et al., 2021; Angelopoulos et al., 2021) and semantic uncertainty intervals for generative adversarial networks (Sankaranarayanan et al., 2022). Sun (2022) provides an extensive survey on distribution-free conformal prediction methods.

**Uncertainty for Image-to-image Regression Tasks**  A work that stands out in this realm is (Angelopoulos et al., 2022b), which experiments with the conformal prediction technique. They evaluate and compare several models that address both the prediction of a value per each pixel, and an accompanying confidence interval that should contain the true value. After calibration, a guarantee is provided such that with high probability the true values of the pixels are indeed within these predicted confidence intervals. They find that quantile regression gives the best results among the four methods considered. As our work addresses the same breed of image-to-image problems, we provide extensive comparison to this work.

## 3 MASK-BASED UNCERTAINTY ESTIMATION

### 3.1 PROBLEM FORMULATION

We consider the problem of recovering an image $y \in [0,1]^n$ from its observations $x \in [0,1]^{n_x}$ and denote by $\hat{y}(x) \in [0,1]^n$ a given estimator of $y$. We define the uncertainty of $\hat{y}$ given $x$ as follow:

$$U(\hat{y}|x) \triangleq \mathbb{E}_{y|x}[d(y,\hat{y})], \tag{1}$$

where $d(\cdot,\cdot)$ denotes an arbitrary distortion measure between the ground truth image $y$ and its estimation $\hat{y}$, and $\mathbb{E}$ is the expectation over the random variable $y|x$. Ideally, we would like to design $\hat{y}$ such that

$$P\Big(U(\hat{y}|x) \leq \alpha\Big) \geq \beta \tag{2}$$

where $0 \leq \alpha$ and $0 \leq \beta \leq 1$ are user-defined, and the probability $P(\cdot)$ is computed over the joint probability of the problem pairs $(y,x)$. However, in practice, condition (2) might not hold. Hence, our goal is to identify areas within $\hat{y}$ which have the most adverse effect on the uncertainty and lead to the violation of (2). Formally, we aim to construct an uncertainty map given by a continuous mask $m \in [0,1]^n$ such that

$$P\Big(\mathbb{E}_{y|x}[d_m(y,\hat{y})] \leq \alpha\Big) \geq \beta \tag{3}$$

where $d_m(y,\hat{y}) \triangleq d(m \odot y, m \odot \hat{y})$. Notice we must avoid trivial and undesired solutions, e.g. $m = 0$, which satisfy (3) but provide no true information regarding the uncertainty. To that end, we

consider solving the following problem

$$\max_m \|m\|_1 \quad \text{subject to} \quad \mathbb{E}_{y|x}[d_m(y, \hat{y})] \leq \alpha,$$
$$\vec{0} \leq m \leq \vec{1}. \tag{P1}$$

In words, we wish to find the minimal masking required to satisfy the distortion constraint. Notice that Problem (P1) does not depend on $\beta$ as we consider an instance regression problem for a given $x$ and an originating $y$. We bring back the probabilistic condition using $\beta$ in the next section, when addressing the whole image manifold.

Under certain conditions, the optimal solution is readily given in a closed-form expression as shown by the next theorem.

**Theorem 1.** *Consider Problem (P1) with the distortion measure $d(y, \hat{y}) = \|y - \hat{y}\|_p^p$ where $p > 1$. Then, for sufficiently small $\alpha$, the optimal mask $m^*$ admits a closed-form solution given by*

$$m^*_{(i)} = \alpha^{\frac{1}{p}} \frac{q_{(i)}}{\left(\sum_{j=1}^n q_{(j)}\right)^{\frac{1}{p}}}.$$

*where for all $j \in [n]$*

$$q_{(j)} \triangleq \frac{1}{\left(\mathbb{E}_{y|x}[|\hat{y}_{(j)} - y_{(j)}|^p]\right)^{1/(p-1)}}.$$

*Proof.* See Appendix C. □

Note that the above implies the following:

**Corollary 1.** *The optimal mask, as defined by the solution of (P1), is directly related to the uncertainty measure in Equation (1) via*

$$U(\hat{y}|x) \propto \sum_{i=1}^n (m^*_{(i)})^{-(p-1)}.$$

*Proof.*

$$\sum_{i=1}^n (m^*_{(i)})^{-(p-1)} \propto \sum_{i=1}^n (q^*_{(i)})^{-(p-1)} = \sum_{i=1}^n \mathbb{E}_{y|x}[|\hat{y}_{(j)} - y_{(j)}|^p] = U(\hat{y}|x).$$

□

Thus, the above establishes a correspondence between the optimal mask and uncertainty as defined in (1). Furthermore, it can be shown that the result of Theorem 1 can be extended to $p = 1$, leading to a binary mask (see Appendix A).

The result of Theorem 1 serves only as a general motivation in our work, since obtaining the optimal mask directly via the above equations is impractical for a number of reasons. First, $\alpha$ may vary depending on the user and the application. Second, for arbitrary distortion measure Problem (P1) does not admit a closed-form solution. Last and most importantly, at inference time the ground truth $y$ is unknown. Hence, our next step is to provide an estimation for the mask which requires no knowledge of $y$, can be computed for any differentiable distortion measure, and is easily adapted for various values of $\alpha$ and $\beta$.

### 3.2 METHOD

Here, we aim to design an estimator for the the uncertainty mask defined by Problem (P1). We start by modeling the mask as a neural network $m_\theta(x, \hat{y})$ with parameters $\theta$, which we refer to as the masking model (see details in Section 4.2). Similar to Problem (P1), we define the following optimization problem

$$\min_\theta \|m_\theta(x, \hat{y}) - \vec{1}\|_2^2 \quad \text{subject to} \quad \mathbb{E}_{y|x}[d_{m_\theta}(y, \hat{y})] \leq \alpha,$$
$$\vec{0} \leq m_\theta(x, \hat{y}) \leq \vec{1}. \tag{P2}$$

Ignoring (for now) the second constraint, the Lagrangian of Problem (P2) is given by

$$L(\theta, \mu) = ||m_\theta(x, \hat{y}) - \overrightarrow{1}||_2^2 + \mu\Big(\mathbb{E}_{y|x}[d_{m_\theta}(y, \hat{y})] - \alpha\Big) \tag{4}$$

where $\mu > 0$ is the dual variable which is considered as an hyperparameter. Given $\mu$, the optimal mask can be found by minimizing $L(\theta, \mu)$ with respect to $\theta$, which is equivalent to minimizing

$$||m_\theta(x, \hat{y}) - \overrightarrow{1}||_2^2 + \mu\mathbb{E}_{y|x}[d_{m_\theta}(y, \hat{y})] \tag{5}$$

since $\alpha$ does not depend on $\theta$. Thus, motivated by the above, we train our masking model using the following loss function:

$$\ell(\mathcal{D}, \theta) = \sum_{(x, \hat{y}, y) \in \mathcal{D}} ||m_\theta(x, \hat{y}) - \overrightarrow{1}||_2^2 + \mu d_{m_\theta}(y, \hat{y}) \tag{6}$$

where $\mathcal{D} \triangleq \{(x_k, \hat{y}_k, y_k)\}_k$ is a dataset of triplets of the degraded input, recovered output and the true image respectively. To enforce that the mask is within $[0, 1]$ we simply clip the output of masking model. Note that the proposed approach facilitates the use of any differentiable distortion measure. Furthermore, at inference time, our masking model provides an approximation of the optimal uncertainty mask – the solution of (P1) – without requiring $y$.

Following the above, notice that the loss function is independent of $\alpha$ and $\beta$, hence, we cannot guarantee that $m_\theta$ satisfies condition (3). To overcome this, we follow previous work on conformal prediction and consider the output of our masking model as an initial estimation of the uncertainty which can be calibrated to obtain strong statistical guarantees. In general, the calibrated mask $m_\lambda$ follows the form

$$m_{\lambda(i)} \triangleq F(m_{\theta(i)}; \lambda), \quad \forall i = 1, ..., n, \tag{7}$$

where $F(\cdot; \lambda)$ is a monotonic non-decreasing function, parameterized by a global scalar which is tuned throughout the calibration process. As our goal is to reduce the distortion below a predefined fixed level, the calibrated mask need to be inversely proportional to the distortion. Since our predicted mask is proportional to the distortion we use the following formula[1],

$$m_{\lambda(i)} = \frac{\lambda}{\epsilon + 1 - m_{\theta(i)}} \quad \forall i = 1, ..., n, \tag{8}$$

which was found empirically to perform well in our experiments. To set the value of $\lambda$ we assume a given calibration dataset $\mathcal{C} = \{(x_k, \hat{y}_k, y_k)\}_k$ and perform the following calibration procedure:

1. For each pair $(x_k, \hat{y}_k)$ we predict a mask $m_k = m_\theta(x_k, \hat{y}_k)$.

2. We define a calibrated mask $m_{\lambda_k}$ according to (8) where $\lambda_k$ is the maximal value such that

$$d\big(m_{\lambda_k} \odot y, m_{\lambda_k} \odot \hat{y}\big) \leq \alpha.$$

3. Given all $\{\lambda_k\}$, we set $\lambda$ to be the $1 - \beta$ quantile of $\{\lambda_k\}$, i.e. the maximal value for which at least $\beta$ fraction of the calibration set satisfies condition (P1).

The final mask, calibrated according to $\alpha$ and $\beta$, is guaranteed to satisfy condition (3) above.

The benefits of our masking approach are as follows. First, it provides a measure of uncertainty which can be easily interpreted. In addition, our training process can accept any distortion measure and learns the relationships and correlations between different pixels in the image. Note thatour model is trained only once, irrespective of the values of $\alpha$ and $\beta$. Thus, our model can be easily adapted for different values of $\alpha$ and $\beta$ via our simple calibration process. Finally, at inference, we can produce our uncertainty map without the knowledge of the ground truth while still providing strong statistical guarantees on our output mask.

---

[1]We add a small value, $\epsilon$, to the denominator for numerical stability and clip the value of $m$ to be in $[0, 1]$ . A formal description of our algorithm can be seen in Appendix B.

## 4 EXPERIMENTS

### 4.1 DATASETS AND TASKS

**Datasets**  Two data-sets are used in our experiments:

Places365 (Zhou et al., 2017): A large collection of 256x256 images from 365 scene categories. We use 1,803,460 images for training and 36,500 images for validation/test.

Rat Astrocyte Cells (Ljosa et al., 2012): A dataset of 1,200 uncompressed images of scanned rat cells of resolution $990 \times 708$. We crop the images into $256 \times 256$ tiles, and randomly split them into train and validation/test sets of sizes 373,744 and 11,621 respectively. The tiles are partially overlapped as we use stride of 32 pixels when cropping the images.

**Tasks**  We consider the following image-to-image tasks:

Completion: Using grayscale version of Places365, we remove a middle vertical and horizontal stripe of 32 pixel width, and aim to reconstruct the missing part.

Super Resolution: We experiment with this task on the two data-sets. The images are scaled down to $64 \times 64$ images and the goal is to reconstruct the original image.

Colorization: We convert the Places365 images to grayscale and aim to recover their colors.

Figure 4 shows a sample input and output for each of the above tasks.

### 4.2 EXPERIMENTAL DETAILS AND SETTINGS

**Image-to-Image Models**  We start by training models for the above three tasks. Note that these models are not intended to be state of the art, but rather used to demonstrate the uncertainty estimation technique proposed in this work. We use the same model architecture for all tasks: an 8 layer U-Net. For each task we train two versions of the network: (i) A simple regressor; and (ii) A conditional GAN, where the generator plays the role of the reconstruction network. For the GAN, the discriminator is implemented as a 4 layer CNN. We use the L1 loss as the objective for the regressor, and augment it with an adversarial loss in the conditional GAN ($\lambda = 20$), as in Isola et al. (2017). All models are trained for ten epochs using the Adam optimizer with a learning rate equal to 1e-5 and a batch size equal to $50$.

**Mask Model**  For our masking model we also use an 8 layer U-Net architecture. This choice was made for simplicity and compatibility with previous work. The input for the mask model is the concatenation of the degraded image and the predicated image on the channel axis. It's output is a mask having the same shape as the predicted image, with values in the range $[0, 1]$. The masking model is trained using  the loss function described in Section 3.2 with $\mu = 2$, learning rate of 1e-5 and a batch size of 25.

**Experiments**  We consider the L1, L2, SSIM and LPIPS distances. We set aside $1,000$ samples from each validation set for calibration and use the remaining samples for evaluation. We demonstrate the flexibility of our approach by conducting experiments on a variety of 12 settings: (i) Image Completion: {Regressor, GAN} × {L1, LPIPS}; (ii) Super Resolution: {Regressor, GAN} × {L1, SSIM}; and (iii) Colorization: {Regressor, GAN} × {L1, L2}.

**Thresholds**  Recall that given a predicted image, our goal is to find a mask that, when applied to both the prediction and the (unknown) reference image, reduces the distortion between them to a predefined threshold with high probability $\beta$. Here we fix $\beta = 0.9$ and choose the threshold to be the 0.1-quantile of each measure computed on a random sample from the validation set, i.e. roughly $10\%$ of the predictions are already considered sufficiently good and do not require masking at all.

### 4.3 COMPETITOR TECHNIQUES

**Quantile – Interval-Based Technique**  We compare our method to the quantile regression option presented in (Angelopoulos et al., 2022b). As described in Section 2, their uncertainty is formulated by pixel-wise confidence intervals, calibrated so as to guarantee that with high probability the true value of the image is indeed within the specified range. While their predicted confidence intervals

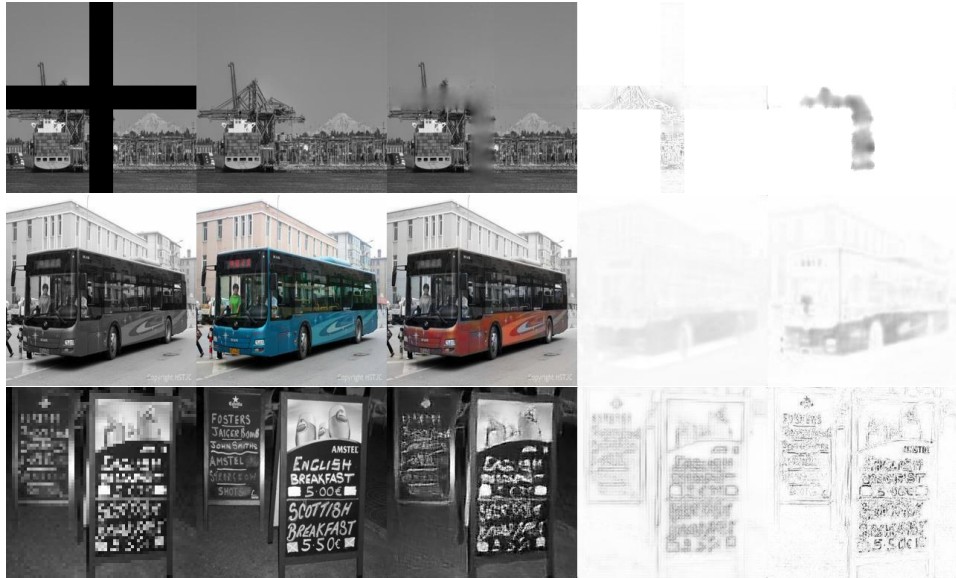

Figure 2: Example of uncertainty masks for image completion (top), colorization (middle) and super resolution (bottom): the images from left to right are the input to the model, the reference image, the output of the model, the calibrated uncertainty mask trained on the L1 loss and the L1 distance heat-map. In the image completion task the bottom left corner is richer in details and thus there is high uncertainty regarding this part in the reconstructed image. For the colorization task, the clothes and the colored area of the bus are the most uncertained regions respectively as these regions can be colorized with a large variaty of colors. In the super resolution task the uncertained regions are around the edges and text while smooth surfaces are more certain parts. Note that the actual error (right most image) is not equivalent to our definition of uncertainty and we only present it as a reference.

are markedly different from the expected distortion we consider, we can use these intervals as a heuristic mask. For completeness, we also report the performance of the quantile regression even when it is less suitable, i.e. when the underlying model is a GAN and when the distortion loss is different from L1. We note again that for the sake of a fair comparison, our implementation of the masking model uses exactly the same architecture as the quantile regressor.

**Opt – Oracle**  We also compare our method with an oracle, denoted Opt, which computes an optimal mask using the following optimization procedure that has access to the reference image $y$:

$$\ell(m) = \min_m ||m - \vec{1}||_2^2 + \mu d_m(y, \hat{y}). \tag{9}$$

This minimization is obtained via gradient descent, using the Adam optimizer with a step-size $0.01$, iterating until the destination distortion value is attained. Clearly, this approach does not require calibration, as the above procedure ensures that all the masked samples are below the target distortion threshold.

## 4.4 RESULTS AND DISCUSSION

We now show a series of results that demonstrate our proposed uncertainty masking approach, and its comparison with Opt and Quantile.[2] We begin with a representative visual illustration of our proposed mask for several test cases, see Figure 2. As can be seen, the mask aligns well with the true prediction error, while still identifying sub-regions of high certainty.

In Figure 3 we explore the performance of the three compared methods with respect to (i) the probabilistic guarantee, (ii) the size of the masks obtained, and (iii) the correlation between mask sizes –

---

[2]Due to space limitations, we show more extensive experimental results in Appendix B, while presenting a selected portion of them here.

| Network | Distance | $\|\mathbf{M}\|$ ($\downarrow$) | | | $\mathbf{C}(\mathbf{M}, \mathbf{D})$ ($\uparrow$) | | $\mathbf{C}(\mathbf{M}, \mathbf{M_{opt}})$ ($\uparrow$) | |
| | | Opt | Ours | Quantile | Ours | Quantile | Ours | Quantile |
|---|---|---|---|---|---|---|---|---|
| Regression | L1 | 0.09 | **0.10** | 0.15 | **0.89** | 0.78 | **0.89** | 0.76 |
| Regression | LPIPS | 0.01 | **0.01** | 0.20 | **0.54** | 0.51 | **0.89** | 0.77 |
| GAN | L1 | 0.09 | **0.09** | 0.14 | **0.95** | 0.85 | **0.94** | 0.80 |
| GAN | LPIPS | 0.01 | **0.01** | 0.08 | **0.31** | 0.24 | **0.50** | 0.23 |
| | | | | Rat Astrocyte Cells | | | | |
| Regression | L1 | 0.24 | **0.26** | 0.28 | **0.99** | 0.54 | **0.95** | 0.88 |
| Regression | SSIM | 0.03 | **0.03** | 0.13 | **0.66** | 0.64 | **0.82** | 0.57 |
| GAN | L1 | 0.26 | **0.30** | 0.40 | **0.94** | 0.63 | **0.80** | 0.72 |
| GAN | SSIM | 0.03 | **0.03** | 0.13 | **0.79** | 0.63 | **0.83** | 0.63 |
| | | | | Places365 | | | | |
| Regression | L1 | 0.30 | **0.36** | 0.39 | **0.99** | 0.97 | **0.95** | 0.94 |
| Regression | SSIM | 0.10 | **0.23** | 0.48 | **0.89** | 0.85 | **0.94** | 0.84 |
| GAN | L1 | 0.37 | **0.38** | 0.47 | **0.97** | 0.81 | **0.95** | 0.67 |
| GAN | SSIM | 0.10 | **0.12** | 0.51 | **0.86** | 0.81 | **0.92** | 0.86 |
| Regression | L1 | 0.27 | **0.37** | 0.40 | **0.68** | 0.43 | **0.57** | 0.46 |
| Regression | L2 | 0.18 | **0.37** | 0.38 | **0.57** | 0.30 | **0.60** | 0.48 |
| GAN | L1 | 0.27 | **0.38** | 0.40 | **0.58** | 0.40 | **0.60** | 0.52 |
| GAN | L2 | 0.18 | **0.36** | 0.38 | **0.42** | 0.28 | **0.59** | 0.49 |

Table 1: Image completion (top), super-resolution (middle) and colorization (bottom), results. $\|\mathbf{M}\|$ indicates average mask size; $\mathbf{C}(\mathbf{M}, \mathbf{D})$ the correlation between predicted mask size and the amount of distortion; and $\mathbf{C}(\mathbf{M}, \mathbf{M_{opt}})$ the correlation between the size of the calibrated mask and the size of the optimal mask. Arrows indicate which direction is better. Best results shown in **blue**. 95% confidence intervals were calculated for all the experiments and are omitted due to lack of space. All presented results were verified for their statistical significance.

Quantile and Ours versus Opt. As can be seen from the top row, all three methods meet the distortion threshold with fewer than $10\%$ exceptions, as required. Naturally, Opt does not have outliers since each mask is optimally calibrated by its computation. The spread of loss values tends to be higher with Quantile, indicating weaker performance. The colorization results, here and below, seem to be more challenging, with a smaller performance increase for our method.

The above probabilistic results are not surprising as they are the outcome of the calibration stage. The main question remaining relates to the size of the mask created so as to meet this threshold distortion. The middle row in Figure 3 presents histograms of these sizes, showing that our approach tends to produce masks that are close in size to those of Opt; while Quantile produces larger, and thus inferior, masked areas. Again, our colorization results show smaller if any improvement.

The bottom row in Figure 3 shows the correlation between mask sizes, and as can be observed, a high such correlation exists between our method and Opt, while Quantile produces less correlated and much heavier masks.

Quantitative results are presented in Table 1 for the completion, super resolution, and colorization tasks. First and foremost, the results obtained by our method surpass those of Quantile across the board. Our method exhibits smaller mask size $\|\mathbf{M}\|$, aligned well with the masks obtained by Opt. Quantile, as expected, produces larger masks. In terms of the correlation between the obtained mask-size and the unmasked distortion value $\mathbf{C}(\mathbf{M}, \mathbf{D})$, our method shows high such agreement, while Quantile lags behind. This correlation indicates a much desired adaptivity of the estimated mask to the complexity of image content and thus to the corresponding uncertainty. The correlation $\mathbf{C}(\mathbf{M}, \mathbf{M_{opt}})$ between Opt's mask size and the size of the mask obtained by our method or Quantile show similar behavior, aligned well with behavior we see in the bottom of Figure 3.

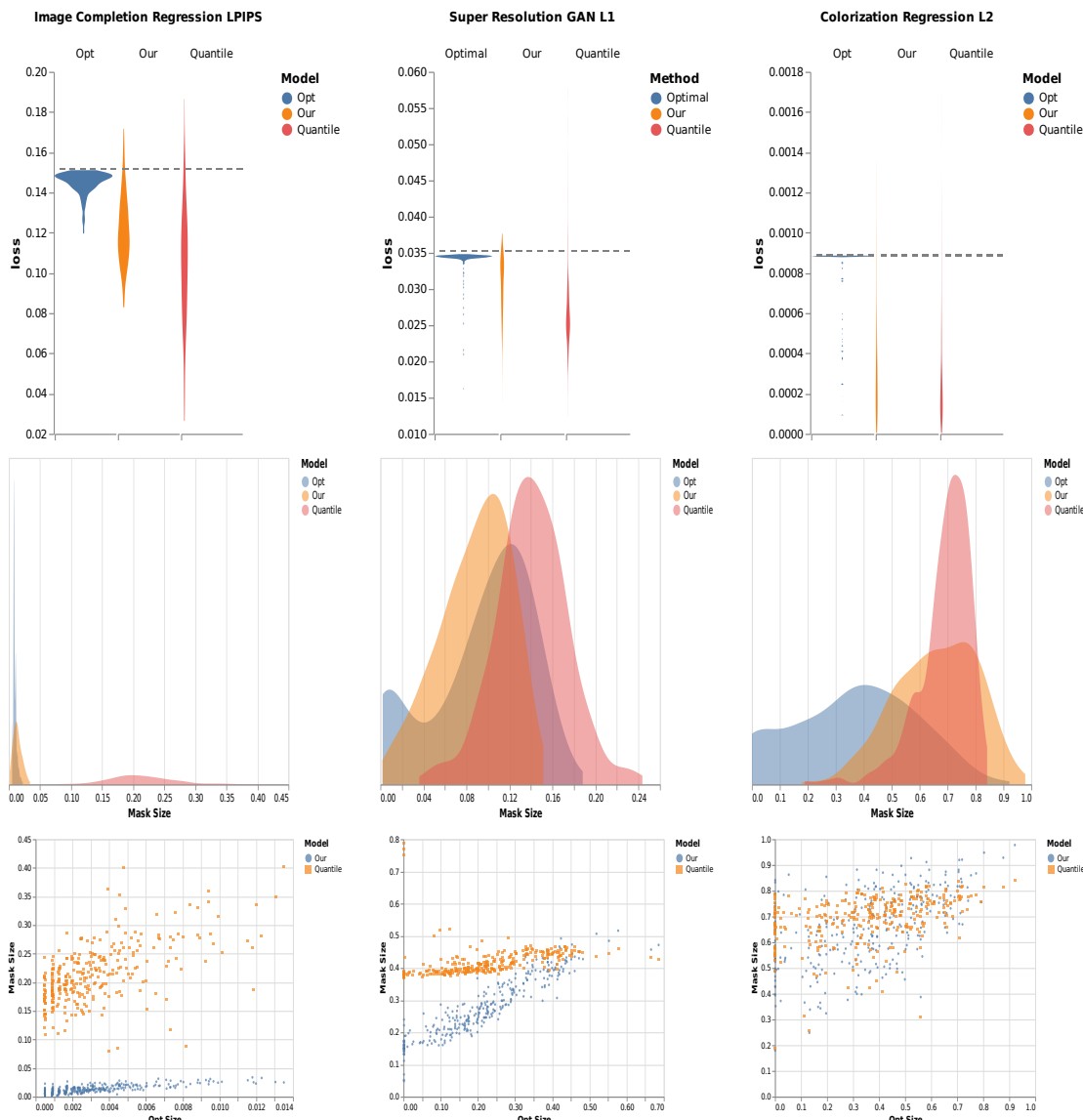

Figure 3: Summary of results for image completion (left), super-resolution (middle) and colorization (right). The top row shows the distribution of distortion values after masking versus the user-chosen threshold. The middle row presents histograms of mask sizes for the three compared methods. The bottom row describes the inter-relation between Opt's mask size and the sizes obtained by Our method and Quantile's.

## 5 CONCLUSIONS

Uncertainty assessment in image-to-image regression problems is a challenging task, due to the implied complexity, the high dimensions involved, and the need to offer an effective and meaningful visualization of the estimated results. This work proposes a novel approach towards these challenges by constructing a continuous mask that highlights the trust-worthy regions in the estimated image. This mask provides a measure of uncertainty accompanied by an accuracy guarantee, stating that with high probability, the distance between the original and the estimated images over the non-masked regions is below a user-specified value. The presented paradigm is flexible, being agnostic to the choice of distance function used and the regression method employed, while yielding masks of small area.

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

## A  TASK ILLUSTRATIONS

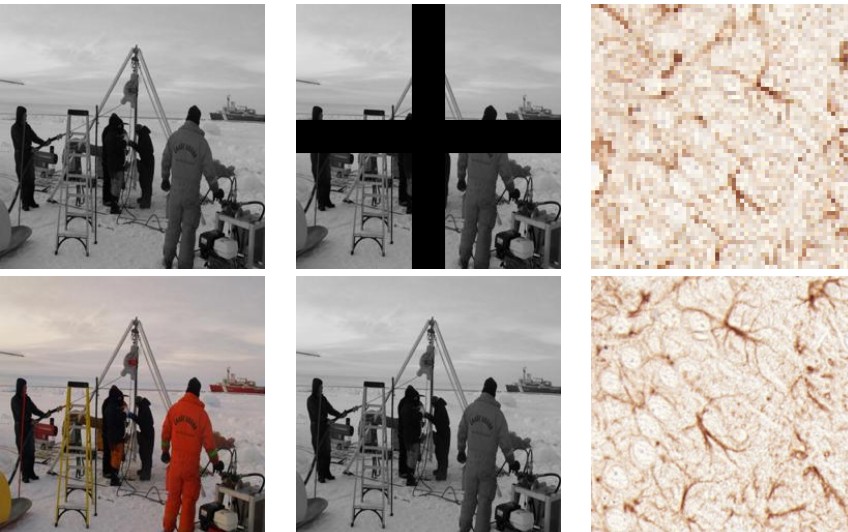

Figure 4:  The three tasks we experimented with:  1) Image Colorization on the left column 2) Gray-scale Image Completion on the middle column, and 3) Super Resolution on the right column.

## B  CALIBRATION ALGORITHM

---
**Algorithm 1** Finding $\lambda_i$

---

```python
def scale(lambda_, mask, eps=1e-6):
    return (mask * lambda_ / (eps + mask * (1 - mask))).clip(0, 1)

def find_lambda(mask, y, y_hat, eps=1e-4):
    l, h, lambda_ = eps, 1 / eps, 0

    while h - l > eps:
        lambda_ = (h + l) / 2

        loss = loss_fn(
            y * scale(a, mask),
            y_hat * scale(a, mask))

        if loss > alpha:
            h = lambda_
        else:
            l = lambda_

    return lambda_
```
---

## C  PROOF OF THEOREM 1

Recall that we have defined the following optimization problem as defining the optimal mask:

$$\max_m \|m\|_1 \quad \text{subject to} \quad \mathbb{E}_{y|x}[d_m(y, \hat{y})] \leq \alpha,$$
$$\vec{0} \leq m \leq \vec{1}. \tag{P1}$$

**Theorem 1:** Consider Problem (P1) with the distortion measure $d(y, \hat{y}) = \|y - \hat{y}\|_p^p$ where $p > 1$. Then, for sufficiently small $\alpha$, the optimal mask $m^*$ admits a closed-form solution given by

$$m_{(i)}^* = \alpha^{\frac{1}{p}} \frac{q_{(i)}}{\left(\sum_{j=1}^n q_{(j)}\right)^{\frac{1}{p}}}.$$

where for all $j \in [n]$

$$q_{(j)} \triangleq \frac{1}{\left(\mathbb{E}_{y|x}[|\hat{y}_{(j)} - y_{(j)}|^p]\right)^{1/(p-1)}}.$$

*Proof.* We start with reformulating Problem (P1) as follows:

$$\min_m \ -\|m\|_1 \quad \text{subject to} \quad \mathbb{E}_{y|x}[d_m(y, \hat{y})] \leq \alpha,$$
$$m_{(i)} \in [0, 1].$$

Next, we define the Lagrangian of the above problem,

$$L(m, \mu) \triangleq -\|m\|_1 + \mu\Big(\mathbb{E}_{y|x}[\|m \odot (y - \hat{y})\|_p^p] - \alpha\Big),$$

where $\mu > 0$ is the dual variable, and we invoke the chosen metric $d(y, \hat{y}) = \|y - \hat{y}\|_p^p$ for $p > 1$. According to the Karush–Kuhn–Tucker (KKT) conditions, the optimal solution $(m^*, \mu^*)$ must satisfy

$$\frac{\partial L}{\partial m_{(i)}} = 0, \quad \forall i = 1, ..., n,$$
$$\frac{\partial L}{\partial \mu} = 0.$$

Furthermore, notice that the Lagrangian is separable with respect to the coordinates of $m$, thus, we can write

$$\frac{\partial L}{\partial m_{(i)}} = \frac{\partial}{\partial m_{(i)}}\Big(-|m_{(i)}| + \mu\big(\mathbb{E}_{y|x}[|m_{(i)} \cdot (y_{(i)} - \hat{y}_{(i)})|^p] - \alpha\big)\Big)$$
$$= \frac{\partial}{\partial m_{(i)}}\Big(-|m_{(i)}| + \mu|m_{(i)}|^p \cdot \mathbb{E}_{y|x}[|y_{(i)} - \hat{y}_{(i)}|^p]\Big)$$
$$= \frac{\partial}{\partial m_{(i)}}\Big(-m_{(i)} + \mu m_{(i)}^p \cdot \mathbb{E}_{y|x}[|y_{(i)} - \hat{y}_{(i)}|^p]\Big)$$
$$= -1 + \mu p \mathbb{E}_{y|x}[|y_{(i)} - \hat{y}_{(i)}|^p] \cdot m_{(i)}^{p-1} = 0.$$

In the above we assume that $m_{(i)} \geq 0$, a fact that should be verified once a solution is formed. Hence, we obtain

$$m_{(i)}^{p-1} = \Big(\mu p \mathbb{E}_{y|x}[|y_{(i)} - \hat{y}_{(i)}|^p]\Big)^{-1}.$$

To simplify our derivation, we define

$$d_{(i)} \triangleq \mathbb{E}_{y|x}[|y_{(i)} - \hat{y}_{(i)}|^p]$$
$$q_{(i)} \triangleq d_{(i)}^{-1/(p-1)}.$$

Thus, we can rewrite the above result equivalently as

$$m_{(i)}^{p-1} = (\mu p d_{(i)})^{-1} \text{ or } m_{(i)} = (\mu p)^{-1/(p-1)} \cdot q_{(i)}.$$

Now, consider the second KKT condition:

$$
\begin{aligned}
\frac{\partial L}{\partial \mu} &= \mathbb{E}_{y|x}[\|m \odot (y - \hat{y})\|_p^p] - \alpha \\
&= \sum_{j=1}^n \mathbb{E}_{y|x}[|m_{(j)} \cdot (y_{(j)} - \hat{y}_{(j)})|^p] - \alpha \\
&= \sum_{j=1}^n m_{(j)}^p d_{(j)} - \alpha \\
&= \sum_{j=1}^n m_{(j)} (\mu p d_{(j)})^{-1} d_{(j)} - \alpha \\
&= (\mu p)^{-1} \sum_{j=1}^n m_{(j)} - \alpha \\
&= (\mu p)^{-1} (\mu p)^{-1/(p-1)} \sum_{j=1}^n q_{(j)} - \alpha = 0. \\
\Rightarrow \mu p &= \left( \frac{\alpha}{\sum_{j=1}^n q_{(j)}} \right)^{-(p-1)/p}.
\end{aligned}
$$

Therefore, the optimal solution $m^*$ is given by

$$
m_{(i)}^* = \alpha^{\frac{1}{p}} \frac{q_{(i)}}{\left( \sum_{j=1}^n q_{(j)} \right)^{\frac{1}{p}}}.
$$

Note that as assumed, the obtained solution $m_{(i)}^*$ is non-negative. Finally, assuming $\alpha$ is sufficiently small such that

$$
\alpha^{\frac{1}{p}} \leq \min_i \left( \frac{q_{(i)}}{\left( \sum_{j=1}^n q_{(j)} \right)^{\frac{1}{p}}} \right)^{-1},
$$

the solution satisfies the second constraint, $m_{(i)}^* \leq 1$, which completes the proof. $\qquad \square$

We now turn to present a closely related result, referring to the case $d(y, \hat{y}) = \|y - \hat{y}\|_1$.

**Theorem 2:** Consider Problem (P1) with the distortion measure $d(y, \hat{y}) = \|y - \hat{y}\|_1$. Then the optimal mask $m^*$ admits a closed-form solution given by

$$
m_{(i)} = \begin{cases} 1, & d_{(i)} < d^* \\ 0, & \text{otherwise.} \end{cases}
$$

where $d_{(i)} \triangleq \mathbb{E}_{y|x}[|y_{(i)} - \hat{y}_{(i)}|]$ and $d^*$ is defined by

$$
\sum_i d_{(i)} \cdot \{d_{(i)} \leq d^*\} = \alpha.
$$

*Proof.* Consider the Lagrangian of the above problem when $d(y, \hat{y}) = \|y - \hat{y}\|_1$:

$$
L(m, \mu) \triangleq -\|m\|_1 + \mu \left( \mathbb{E}_{y|x}[\|m \odot (y - \hat{y})\|_1] - \alpha \right),
$$

where $\mu > 0$ is the dual variable. As before, the problem is separable with respect to the coordinates of $m$, leading to the following one-dimensional problem

$$
\min_{0 \leq m(i) \leq 1} -m_{(i)} + \mu m_{(i)} d_{(i)} = \min_{0 \leq m(i) \leq 1} m_{(i)} \left( \mu d_{(i)} - 1 \right),
$$

where $d_{(i)} \triangleq \mathbb{E}_{y|x}[|y_{(i)} - \hat{y}_{(i)}|]$. Thus,

$$
m_{(i)} = \begin{cases} 1, & d_{(i)} < \frac{1}{\mu} \\ 0, & \text{otherwise.} \end{cases}
$$

Observe that this result is intuitive in retrospect, as smaller distances $d_{(i)}$ correspond to un-masked areas. In addition, regardless of $\mu$, the choice of the mask value is dictated by the distance, a fact that we leverage hereafter. Recall that the solution should satisfy

$$\sum_{i}^{n} m_{(i)} d_{(i)} = \alpha,$$

where $m_{(i)}$ are 1-es for the smaller values of $d_{(i)}$. Thus,

$$\sum_{i} m_{(i)} d_{(i)} = \sum_{i:d_{(i)}<d^*} d_{(i)} = \alpha.$$

Hence, the above provides a definition for the threshold value $d^*$, and $\mu = 1/d^*$, which completes the proof. □

## D  MORE RESULTS

In this appendix we bring the results corresponding to all 12 settings explored in our work, corresponding to three inverse problems, two regression techniques and two metrics per each, along the following breakdown:

- Image Completion: {Regressor, GAN} $\times$ {L1, LPIPS};
- Super Resolution: {Regressor, GAN} $\times$ {L1, SSIM}; and
- Colorization: {Regressor, GAN} $\times$ {L1, L2}.

We start in Figures 5-7 with the obtained distributions of masked distortion values for Opt, Ours and Quantile. The goal here is to show that all three methods meet the required distortion threshold with exceptions that do not surpass the destination probability $\beta = 0.1$.

Figures 8-10 present histograms of the obtained mask sizes for the three methods. The goal is to get minimal areas in these masks, so as to keep most of the image content unmasked.

Figures 11-13 conclude these results by bringing graphs showing the inter-relation between the Opt mask size and the ones given by Ours and Quantile.

Analysis and conclusions drawn from these graphs are brought in the experimental part of the paper.

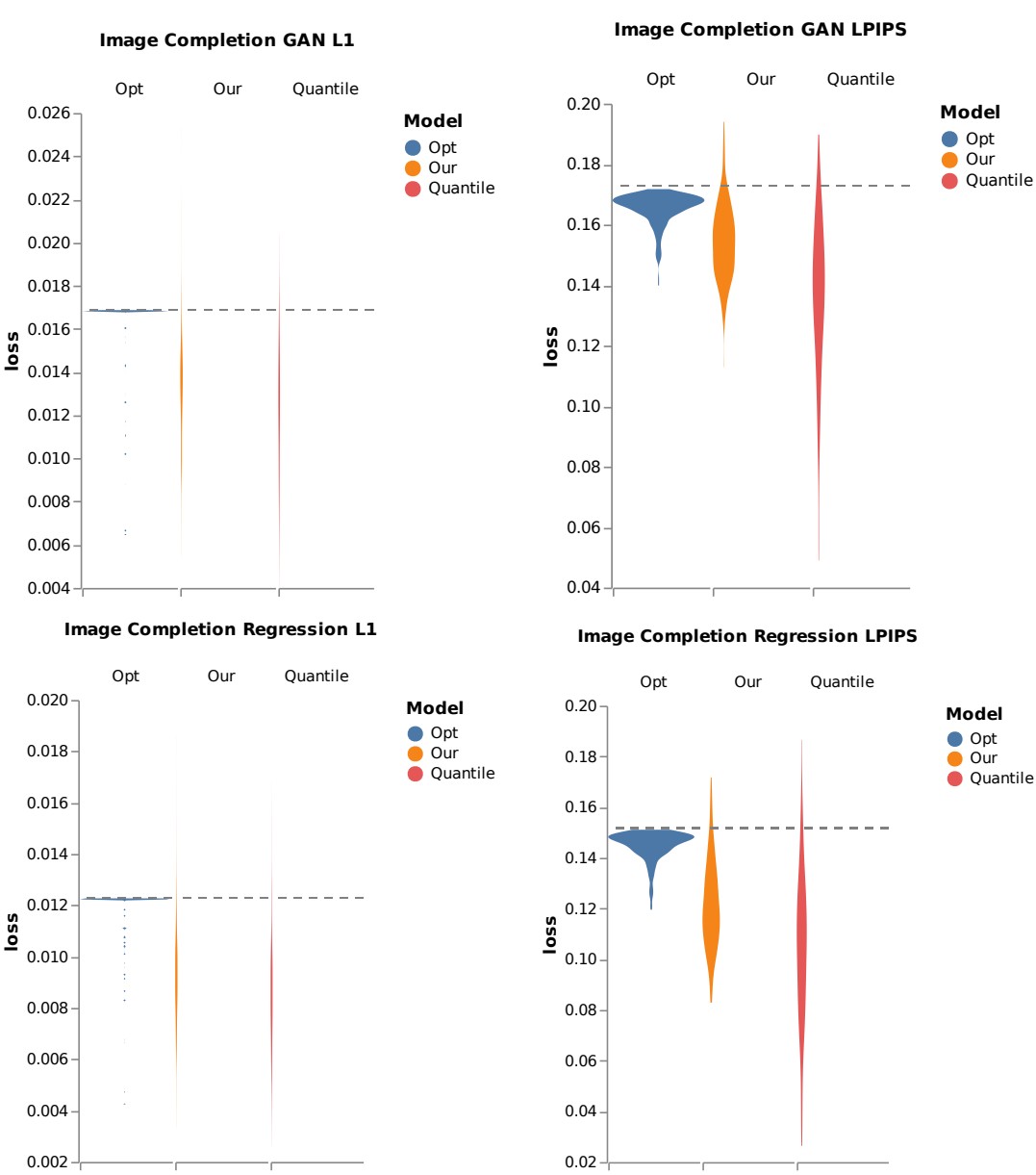

Figure 5: Image Completion - The distribution of the masked distortion values versus the chosen threshold (shown as a horizontal dashed line) for the three tested methods - Opt, Ours and Quantile.

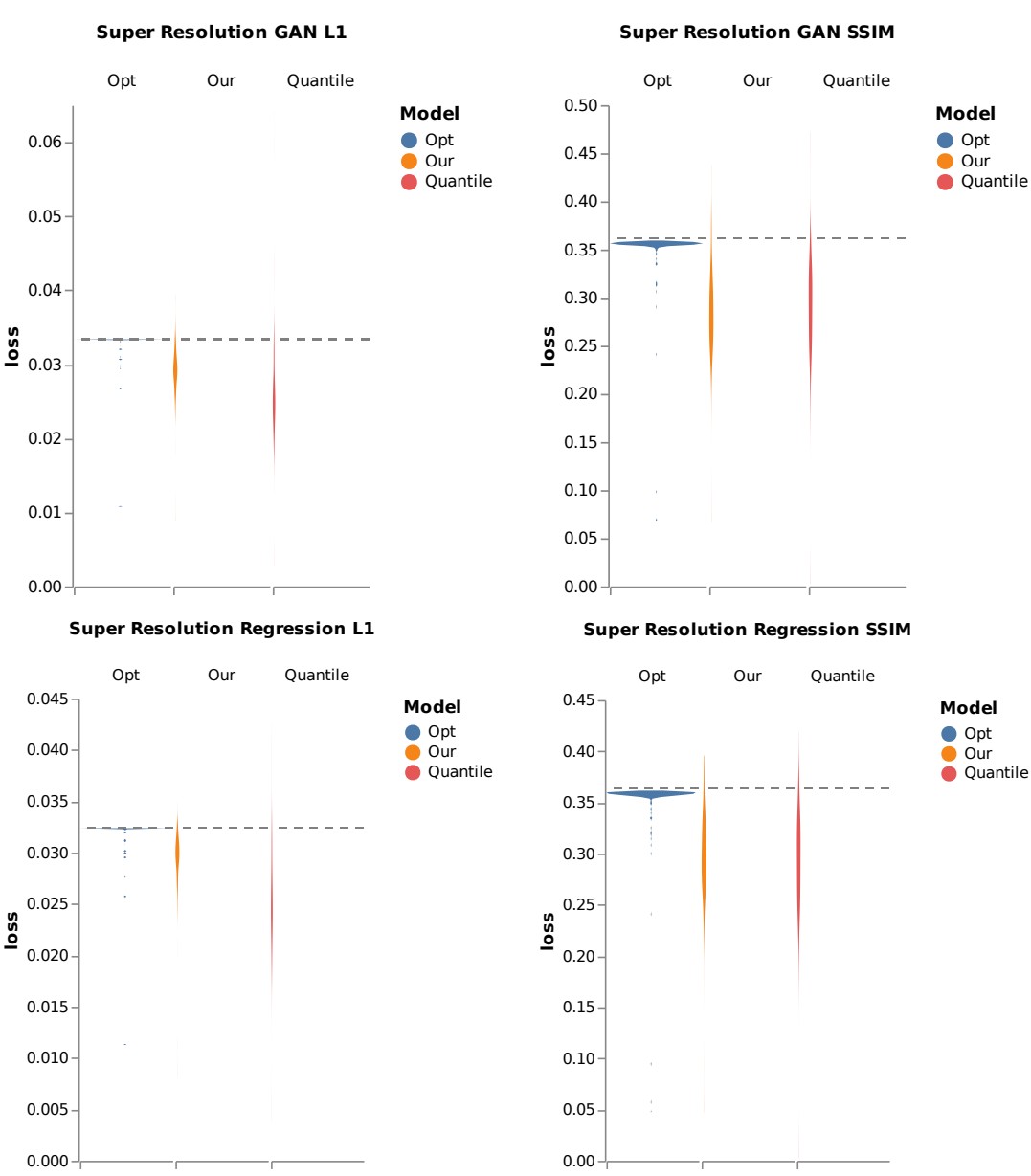

Figure 6: Super Resolution - The distribution of the masked distortion values versus the chosen threshold (shown as a horizontal dashed line) for the three tested methods - Opt, Ours and Quantile.

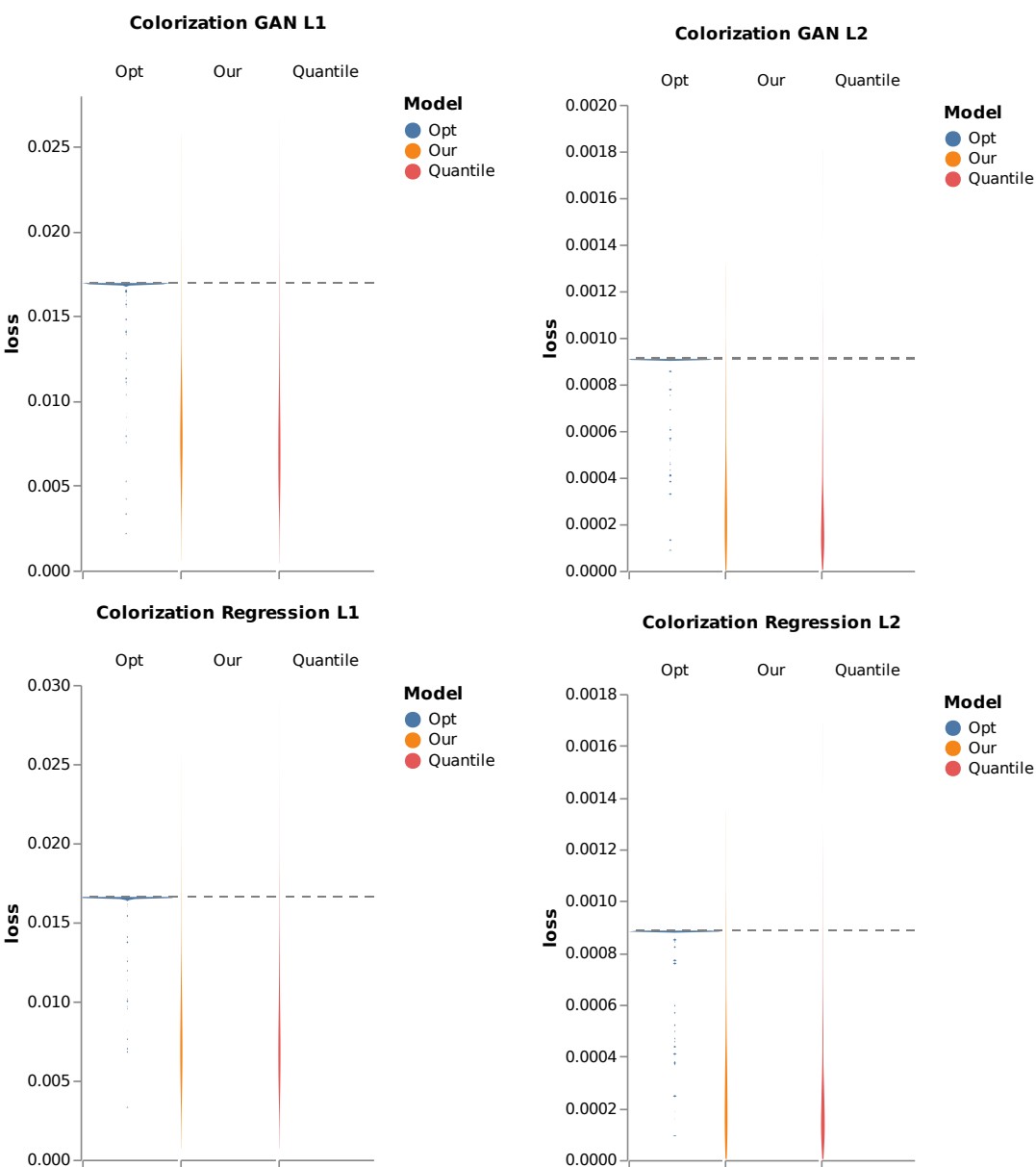

Figure 7: Colorization - The distribution of the masked distortion values versus the chosen threshold (shown as a horizontal dashed line) for the three tested methods - Opt, Ours and Quantile.

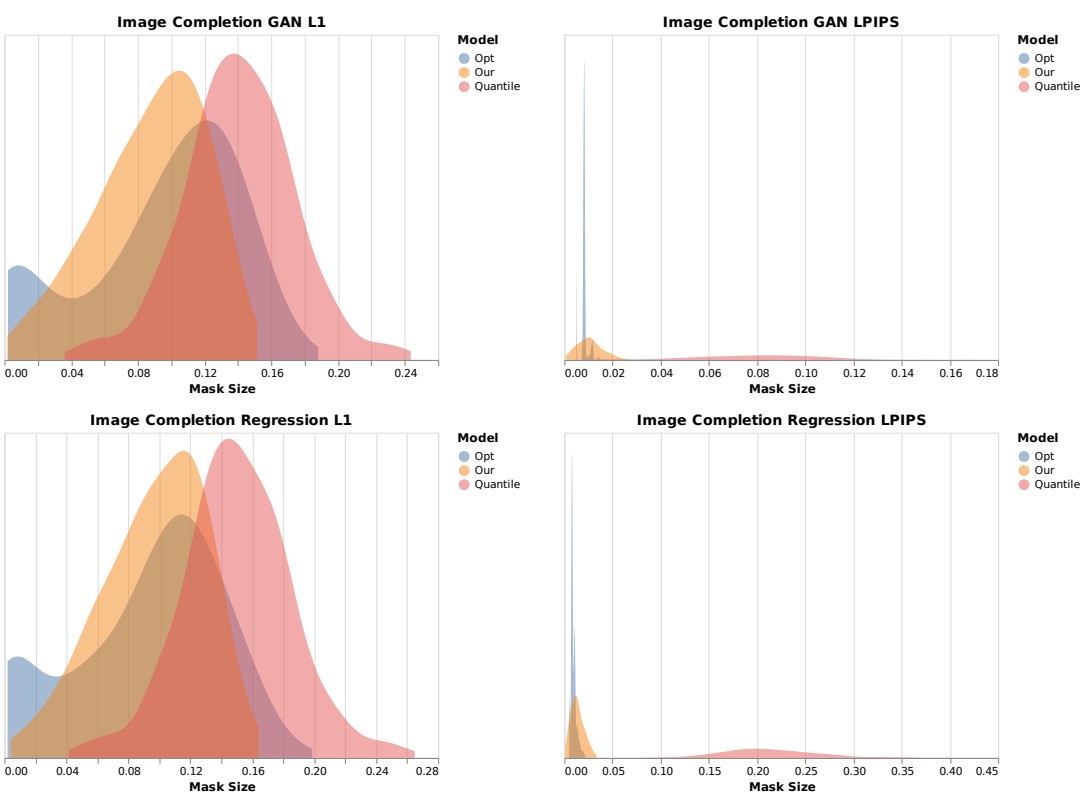

Figure 8: Image Completion - Histograms of the calibrated mask sizes for the three tested methods - Opt, Ours and Quantile.

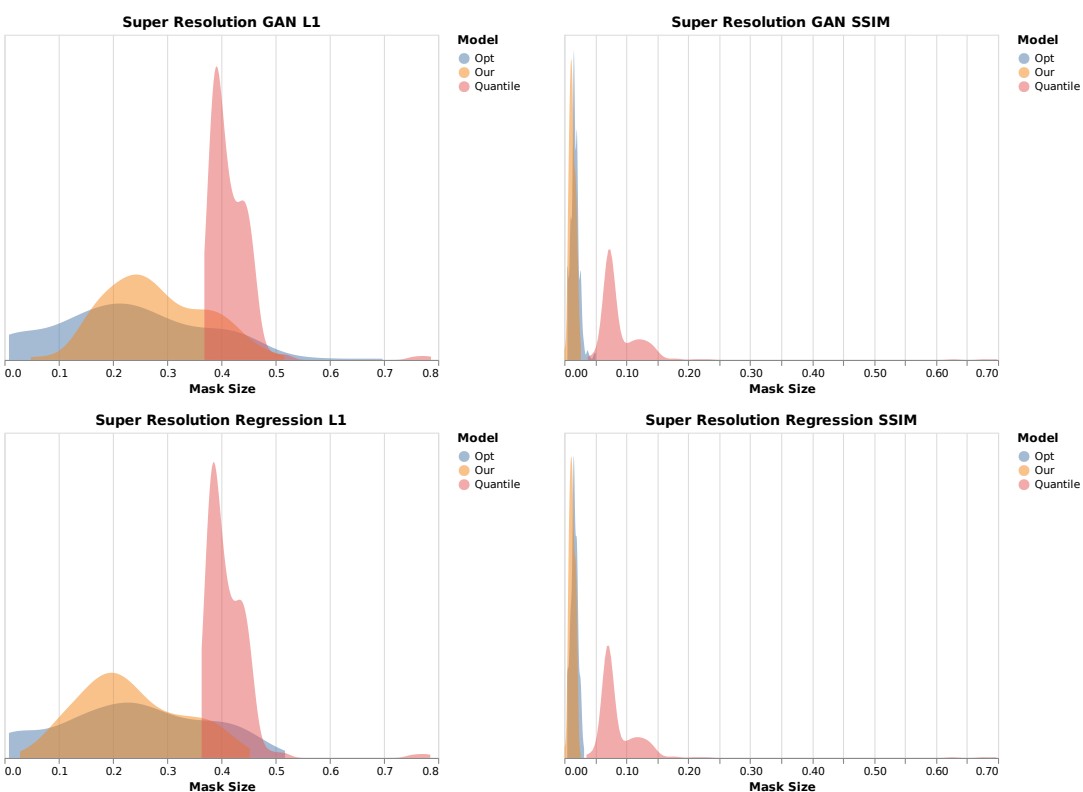

Figure 9: Super Resolution - Histograms of the calibrated mask sizes for the three tested methods - Opt, Ours and Quantile.

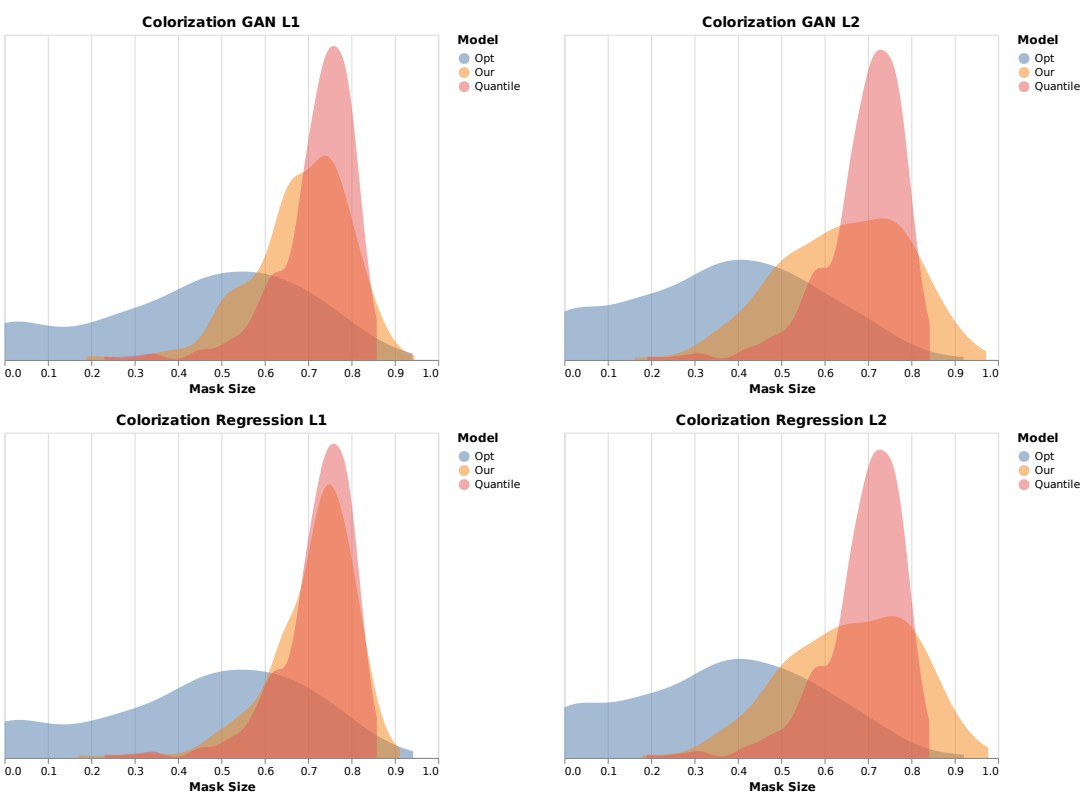

Figure 10: Colorization - Histograms of the calibrated mask sizes for the three tested methods - Opt, Ours and Quantile.

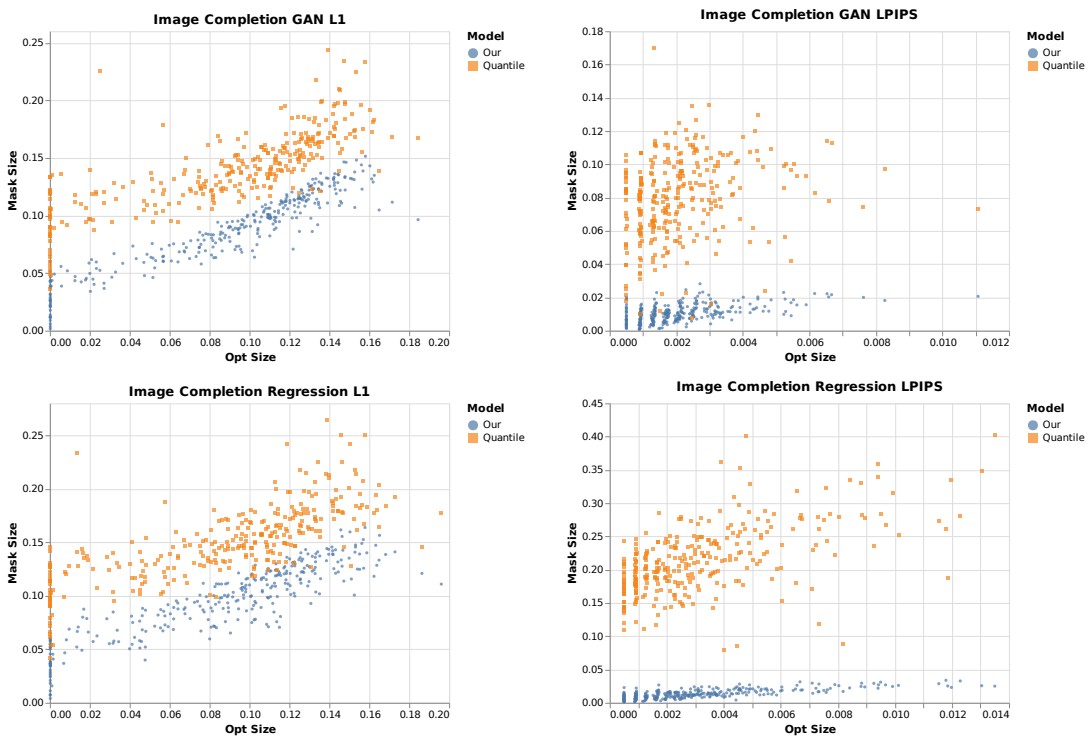

Figure 11: Image Completion - The correlation between the Ours and Quantile mask sizes versus Opt's mask size.

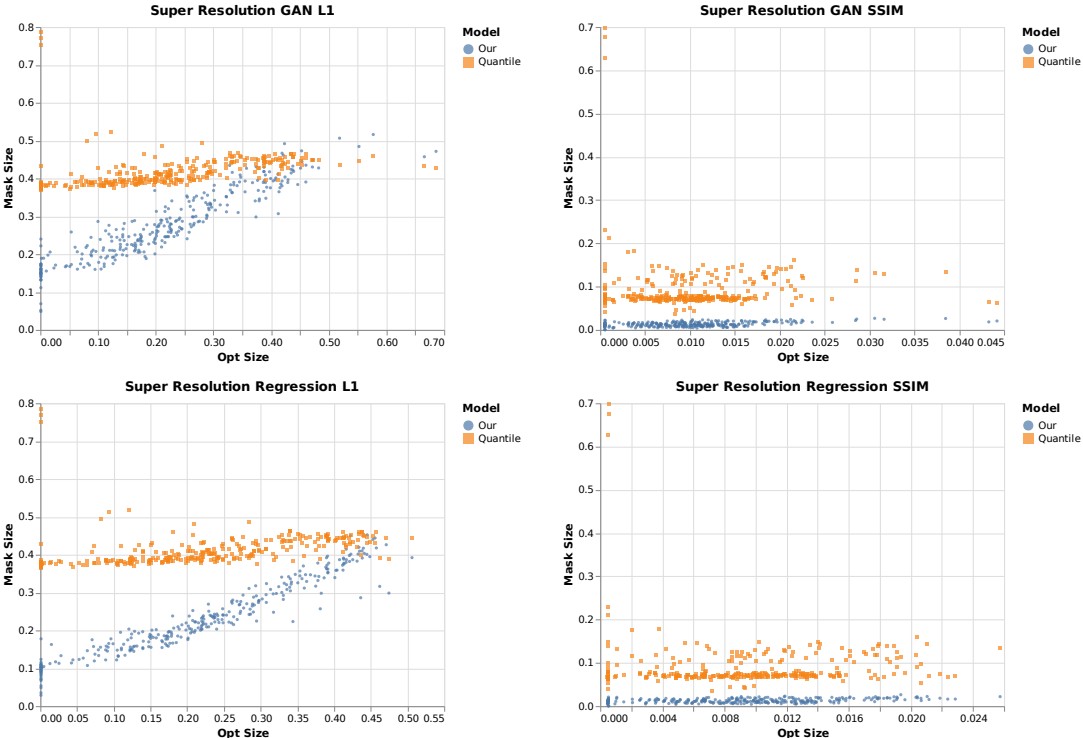

Figure 12: Super Resolution - The correlation between the Ours and Quantile mask sizes versus Opt's mask size.

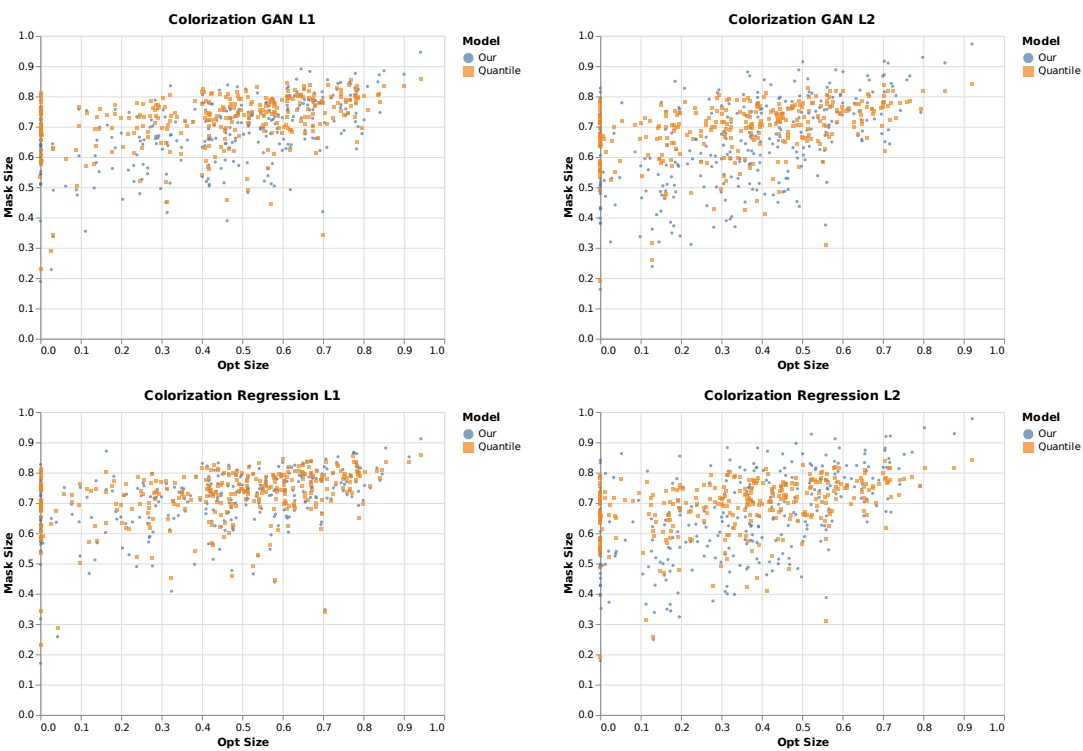

Figure 13: Colorization - The correlation between the Ours and Quantile mask sizes versus Opt's mask size.

