# OpenReview forum: "What's Behind the Mask: Estimating Uncertainty in Image-to-Image Problems"
_ICLR.cc/2023/Conference — Submitted to ICLR 2023_

### Official Review · Reviewer_GH3K · 2022-10-25

**Confidence:** 3
**Correctness:** 3
**Technical Novelty And Significance:** 3
**Empirical Novelty And Significance:** 3
**Recommendation:** 6

**Clarity, Quality, Novelty And Reproducibility:**

This paper is overall well-written.
The method of the work is well described and theoretically solid, with the correspondence between the mask and uncertainty being proved.
The method utilizes continuous masks that can work as uncertainty map as well and can be applied to any image-to-image network and distance function, which is flexible and somewhat novel.


**Strength And Weaknesses:**

Strength
+ theoretically proved that the uncertainty mask is with an accuracy guarantee, and the user can set the threshold to control the ratio of trust-worthy regions.
+ experiments done on three image-to-image tasks prove the adaptability of the proposed method.

Weaknesses
+ in the mask of the colorization task, there is some inexplicable circular artifact pattern, which would depreciate the uncertainty mask.
+ in table 1, the proposed method produces comparable results to that of Opt in some settings, it’s not mentioned whether significant tests are performed or not.  It’s said that a smaller average mask size ||M|| indicates better performance, why Opt’s results are not the best?


**Summary Of The Paper:**

This paper proposes an approach to uncertainty estimation for image-to-image tasks based on continuous masking. The approach can work with arbitrary networks, and distance metrics between images and is guaranteed to yield distances within a small threshold with high probability. Experiments are conducted on image colorization, image completion, and super-resolution tasks.

**Summary Of The Review:**

The paper presents a novel method that connects uncertainty estimation with masks. The method is highly flexible and has a theoretical guarantee. In the experiments, the method is tested on three different image-to-image tasks, which are comprehensive. While the results and comparison are not very intuitive and clear, which can be further improved.

---

> ### Author Response · Authors · 2022-11-17
> **Response to reviewer GH3K**
>
> We thank the reviewer for their comments and suggestions.
>
> **in the mask of the colorization task, there is some inexplicable circular artifact pattern, which would depreciate the uncertainty mask.**
>
> We revisited our colorization experiments and improved them by better training our models, and the above mentioned artifacts are totally removed. We report the new results in the revised version of the paper, both quantitatively and qualitatively.
>
> **in table 1, the proposed method produces comparable results to that of Opt in some settings, it’s not mentioned whether significant tests are performed or not. It’s said that a smaller average mask size ||M|| indicates better performance, why Opt’s results are not the best?**
>
> We calculated 95% confidence intervals for all the values in the Table and omitted them due to lack of space. All presented results were verified for their statistical significance.

---

### Official Review · Reviewer_xoZ2 · 2022-10-26

**Confidence:** 4
**Correctness:** 3
**Technical Novelty And Significance:** 3
**Empirical Novelty And Significance:** 2
**Recommendation:** 5

**Clarity, Quality, Novelty And Reproducibility:**

The paper is largely clearly written, except for a few typos. The idea of minimizing a mask that captures distortion as a measure of uncertainty is certainly novel but if this exactly captures meaningful uncertainty is something that remains unclear.

**Strength And Weaknesses:**

**Strengths:**

* The problem formulations are clearly presented with elegant theoretical justifications. The proof of Theorem 1 solved using a dual formulation of P1 with KKT conditions look correct. The limitations of formulating the problem P1 are discussed clearly, and the motivation for simplifying it to P2 is reasonable.

* The overall task of estimating uncertain regions in image-to-image tasks is interesting, and the diversity of tasks addressed in this work is comprehensive.

* The resulting method is simple and looks like can be easily integrated into other image-to-image tasks.

* Experimental results show the proposed method to consistently perform better than the baseline. There are some issues with the evaluation itself which is discussed further below.

**Weaknesses:**

* **Simplification from P1 to P2**: The simplification from P1 to P2, and finally the objective in Eq. 6 used to train the model assumes a series of strong simplifications. While I see the need for making these simplifications; for instance, going from P1 to P2 removes the reliance on ground truth masks at inference, the implications of making these simplifications remains unclear. One could ideally arrive at the same objective in Eq. 6 without theorising about P1/P2, as it is in the end trying to minimize the span of a mask and distortion. I appreciate the authors presenting theorems and proofs in Appendix A, but I am not entirely convinced this to be useful in the end. I would like clarifications on how the final objective in Eq. 6 still is a reasonable solution to P1.

* **Distortion measures**: The modularity of the proposed method, that it is not tied to specific distance or distortion measures is appealing. However, there is no specification of the distortion measures at all (or that I could easily find). In the end, for the three experiments which are the distortion measures used? Is there a difference in which distortion measure $d(\cdot)$ is used for the three tasks.

* **Loss function in Eq. 6**: During training the loss function in Eq. 6 still uses the ground truth. If this method were to be an unsupervised one is there a further simplification of Eq. 6 that could remove the reliance on ground truth? For instance, the distortion measure could instead be based on some variation of pixel-level entropy of the reconstructed data.

* **Pixel-wise uncertainty measures**: One other simplification used in this work is that of the pixels being independent and their uncertainty measures are estimated/modelled at the pixel level (please correct me if I have misunderstood this part). If this is the case, how reliable are these uncertainty masks? At least from the visualizations in Fig. 3 the mask seems largely spatially correlated. How is this ensured in the model?

* **Mask as a neural network**: In Sec 3.2 the authors state _modelling the mask as a neural network_. What does this exactly mean? The neural network $m_\theta(x,\hat{y})$ takes the degraded input and the reconstructed output and predicts the mask, as a pixel grid with [0,1] values? This is not entirely clear and also the neural network itself is later described as the mask model (implemented as a U-net); are these two the same? If the mask model is indeed a U-net what are the inputs to it? Are the degraded input and reconstruction both provided as the input?

* **Mask-size as a measure of uncertainty**: One of the main concerns with the evaluations reported in this work is using mask size as measure of uncertainty or the correlation of mask size with distortion. This is firstly an unconventional way of assessing uncertainty; perhaps also due to the lack of clarity on what distortion measures are used it seems to be not the most obvious way of representing uncertain regions. Additional results are reported in Fig. 4 but are not discussed in the context of uncertainty assessment, making it harder to wrap them in understanding the proposed work.


**Summary Of The Paper:**

Quantifying uncertain regions when performing image-to-image tasks such as inpainting or super resolution can be useful in identifying low confidence regions. This work presents a mask minimization approach that strives to obtain the smallest regions that correspond to the the most uncertain regions, as defined by user defined probabilistic thresholds. The method is implemented as a neural network and is modular in which distance- or distortion measures can be used in assessing the quality of masks. Experiments are reported on three image-to-image tasks : inpainting, colorization, super-resolution on two datasets using multiple distance measures. The experimental evaluation on two datasets are compared with an oracle based method (Opt) and a related conformal prediction method (Quantile), showing improvements compared to the latter.


**Summary Of The Review:**

A method to estimate uncertainty in image-to-image tasks is presented within a theoretical formulation. However, the problem formulations are simplified extensively to obtain the final objectives; implications of these simplifications are not explained in relation to the original problem formulation. Assessing uncertainty in the experiments is done using mask size as a proxy, which is unconventional and does not immediately translate into established notions of uncertainty estimation methods.

---

> ### Author Response · Authors · 2022-11-17
> **Response to reviewer xoZ2**
>
> We thank the reviewer for their comments and suggestions.
>
> **Simplification from P1 to P2**
>
> Equation 6 poses an optimization procedure for the learned mask m with two forces: (1) Push the masked values to 1 so as to preserve most of the image content; and (2) reduce the masked distances between the true and the estimated images. If this expression would have been summed over all possible triplet values $(x,{\hat y}$ and $y$, this would be exactly equivalent to P2 (for properly chosen parameters $\mu$). Thus, the transitions we present start with intuitively agreed-upon objectives, and modify them eventually to a tractable optimization problem. Please note that for an equivalence as claimed above, we should hold many triplet instances where the same x gets different origin images y, and vice-versa. While the later is easily obtained, the former is impossible to achieve, and thus we rely on the later calibration to accommodate for this lack of balance in the data.
>
> **Distortion measures**
>
> Please note that our experiments included the use of L1, L2, LPIPS and SSIM distance functions.
>
>
> **Loss function in Eq. 6**
>
> This is indeed an intriguing question, and one that puzzled us as well. True - the problem posed in equation 6 leads to a supervised learning paradigm. In our follow up work we are considering unsupervised alternatives based on diffusion models, which can provide a group of estimations $\hat{y}$, serving as samples from the posterior.
>
> **Pixel-wise uncertainty measures**
>
> Thank you for this question. The proposed method can take spatial correlations and statistical dependencies between pixels into account. This is done solely by the distortion function being used. We added a clarification about this important point in the revised paper. As for the spatially correlated masked obtained, this could be explained by the true uncertainty which is also spatially correlated.
>
> **Mask as a neural network**
>
> Sorry for the lack of clarity. The neural network does exactly as described above - taking the degraded input and the reconstructed result and producing as an output a 2D array (having the image size) of the mask values. We improved our explanation of this point in the revised paper.
>
> **Mask-size as a measure of uncertainty**
>
> Please recall that after calibration we guarantee that the masked distance is below a pre-specified threshold. However, this statement by itself is meaningless, since we could easily obtain such a result with a zero mask. Thus, we accompany our tests with the mask size, a scalar value that describes the amount of unmasked region, where 0 stands for no masking at all (the best possible case).  Note that the mask size parallels the average length of the confidence intervals in earlier work.

---

> > ### Comment · Reviewer_xoZ2 · 2022-12-12
> > **Response to author response**
> >
> > I thank the authors for their clarifications. However, my (main) concern about using mask-size as a measure of uncertainty still persists. Also, taking comments from other reviewers also into consideration I will keep my score.

---

### Official Review · Reviewer_DEta · 2022-10-30

**Confidence:** 4
**Correctness:** 3
**Technical Novelty And Significance:** 3
**Empirical Novelty And Significance:** 2
**Recommendation:** 3

**Clarity, Quality, Novelty And Reproducibility:**

     The writing is pretty clear. Authors are did a great job there. The only
     missing parts are some details that confuses me about the equations, i.e.,
     Equation 8 and what it implies.

     The work seems like novel. However, for the comparison, I believe there are
     other works that aimed to predict the expected error or a measure that
     relates to expected error in image-to-image networks. A simple example is
     the aleatoric uncertainty formulation of Gal et al. At the output, they
     simply put a variance that would modulate itself for erroneous
     predictions. The variance map can be used as the map authors produce here.

     The work seems quite reproducible. Authors did a great job there as well.

     The quality of the work is high in most parts. I believe the positioning of
     the article can be better done. Moreover, the comparisons are very
     limited. When we look at this as predicting expected error, I believe
     different comparisons can be considered. I suggest authors to view their
     method from that angle.

**Strength And Weaknesses:**

     Strengths:
     1. The proposed method is a post-hoc analysis technique and, therefore, it
        can be applied to any pre-trained network. This is a clear strength.
     2. Predicted masks are not as good as the upper bound, but they are much
        better than the alternative technique.
     3. Authors demonstrate the method on three different tasks colorization,
        super-resolution and image completion.
     4. Authors demonstrate their method on biological imaging, where
        uncertainty quantification may indeed be important.

     Weaknesses:
     1. The proposed technique does not seem to predict uncertainty. To the best
        of my understanding, it predicts where the original model is making
        errors. An easy way to see this would be to consider a point where the
        model is confident but inaccurate, i.e., a bias. The predicted mask
        should cover this point, even though it is not an uncertain prediction
        of the model. The problem is actually in the definition of
        ``uncertainty''. Equation 1 (or 3) do not necessarily only correspond to
        uncertainty. It corresponds to expected error for a sample. It also
        includes the contribution of bias. In this regard, I think the
        uncertainty positioning may not be very accurate. Consequently, the
        comparison with a method that computes confidence intervals may not be
        appropriate. Here, I should note that predicting expected error or the error is not a
        bad target. However, the difference between this and uncertainty
        estimation should probably be made clear.

     2. I am not sure about the contribution of the corollary 1. The result is
        not very surprising in my opinion. The mask that aims to minimize the
        ``uncertainty'' is bound to be related to the ``uncertainty''.

     3. It is unclear how authors estimate the expectation in Equations 4
        or 5. This is over y|x variable. However, in Equation 6, they drop the
        expectation and simply take only one sample to compute this
        expectation. Using this I am assuming the model learns to predict the
        error and not the expected error. As a result, the model may not be able
        to identify an uncertain prediction at a pixel for a given sample since
        for that pixel, there may be only 1 output that happens to be close to
        the ground truth.

     4. Equation 8 yields a suspicious behavior. When m_{\theta} = 0,
        m_{\lambda} = \lambda. However, when m_{\theta} = 1, m_{\lambda} =
        \infty. It is unclear how authors deal with this. Furthermore, the
        intuition of this specific calibration form is unclear.

**Summary Of The Paper:**

     Authors present a new method for estimating the uncertainty in
     image-to-image problems. Given a model that is already trained to do
     image-to-image prediction, authors train a model that predicts a mask that
     only retains the accurate areas in the prediction. The inaccurate areas are
     masked out, therefore reducing the overall prediction error. The mask is
     said to retain ``certain'' regions in the prediction.

     Authors compare their method with two alternatives. First, is a method that
     predicts confidence intervals. The confidence intervals are modified by the
     authors of this work to generate a mask. The resulting mask is compared to
     that given by the method here. The second alternative is an upper-bound. A
     method that is aware of the prediction and the ground-truth.

     Results show that the proposed method is indeed better than the
     alternative and close to the upper-bound.

**Summary Of The Review:**

     The work is interesting but there are some drawbacks.

     The most important drawback is that authors effectively present a method
     that predicts error maps rather than uncertainty in the prediction.

---

> ### Author Response · Authors · 2022-11-17
> **Response to reviewer DEta**
>
> We thank the reviewer for their comments and suggestions, and we do hope that our explanations clarified better our contribution.
>
> **The proposed technique does not seem to predict uncertainty.
> To the best of my understanding, it predicts where the original model is making errors.
> An easy way to see this would be to consider a point where the model is confident but inaccurate, i.e., a bias.
> The predicted mask should cover this point, even though it is not an uncertain prediction of the model.
> The problem is actually in the definition of ``uncertainty''.
> Equation 1 (or 3) do not necessarily only correspond to uncertainty.
> It corresponds to expected error for a sample. It also includes the contribution of bias.
> In this regard, I think the uncertainty positioning may not be very accurate.
> Consequently, the comparison with a method that computes confidence intervals may not be appropriate.
> Here, I should note that predicting expected error or the error is not a bad target.
> However, the difference between this and uncertainty estimation should probably be made clear.**
>
> Our definition of uncertainty (Eq. 1) embarks from the posterior distribution, $P(y|x)$, evaluating the distance of each candidate solution $\hat{y}$ to the true image $y$, and averaging these distances. The added mask aims to identify regions in the recovery that are “not to be trusted”, thus bringing the above expected masked distance to be below some predetermined threshold. If our estimator would be biased, as suggested, this necessarily influences the uncertainty measure proposed, and thus the mask is expected to detect such regions and mask them out - as should be the case. Therefore, we do believe that our definitions follow an uncertainty prediction rationale.
>
> The comparison with previous work was made mainly to demonstrate the flexibility of our proposed solution where previous work can not be applied directly. We made our best effort to make this comparison as appropriate as possible.
>
> **I am not sure about the contribution of the corollary 1.
> The result is not very surprising in my opinion.
> The mask that aims to minimize the "uncertainty" is bound to be related to the "uncertainty".**
>
> As stated right after Theorem 1 and Corollary 1, these presented results serve as a general motivation of our work, posing a sanity verification for the well-posedness of the defined optimization goal and its implications. Furthermore, the obtained result identifies the exact mathematical relation between the mask and the pixel-wise expected error over the conditional distribution, a valuable asset on its  own, as it supports our continuous valued mask.
>
> **It is unclear how authors estimate the expectation in Equations 4 or 5.
> This is over $y|x$ variable.
> However, in Equation 6, they drop the expectation and simply take only one sample to compute this expectation.
> Using this I am assuming the model learns to predict the error and not the expected error.
> As a result, the model may not be able to identify an uncertain prediction at a pixel for a given sample since for that pixel, there may be only 1 output that happens to be close to the ground truth.**
>
> True, equations 4 and 5 are not computable due to their reliance on the unknown posterior probability $P(y|x)$. The migration to equation 6 makes the optimization tractable, by sweeping through the triplets $x, \hat{y}$, and $y$ - measurement, estimated result and true image, respectively. In a perfect world in which for each $x$ we would have gotten many candidate original images to match it, this change would have been valid, as it preserves the essence of the earlier expectation. As we are short on such data, just as noted, we are likely to get a sub-optimal mask, but please recall that it is later adjusted by the calibration procedure.
>
> **Equation 8 yields a suspicious behavior. When $m_{\theta} = 0$, $m_{\lambda} = \lambda$.
> However, when $m_{\theta} = 1, m_{\lambda} =\infty$.
> It is unclear how authors deal with this. Furthermore, the intuition of this specific calibration form is unclear.**
>
> This is true, and we have fixed the formula to avoid this difficulty by adding a limiting $\epsilon$ in the denominator. In addition, we clip the resulting mask to be in the [0, 1] interval.
>
> **The writing is pretty clear. Authors are did a great job there. The only missing parts are some details that confuses me about the equations, i.e., Equation 8 and what it implies.**
>
> Admittedly, equation 8 presents an arbitrary look-up-table transition of the mask values, which was empirically found to perform better (after calibration). A better, learnable, approach would have led to improved results, but we chose not to pursue such an option in this work.

---

### Official Review · Reviewer_pBvq · 2022-11-02

**Confidence:** 4
**Correctness:** 3
**Technical Novelty And Significance:** 3
**Empirical Novelty And Significance:** 2
**Recommendation:** 5

**Clarity, Quality, Novelty And Reproducibility:**

The clarity and quality of this paper seem good, the proposed idea is somewhat interesting, and the reproducibility is reasonable.

**Strength And Weaknesses:**

Strength:
1. The proposed method introduced a continuous mask to replace the previous binary mask for better modeling the uncertainty between the ground truth image and the reconstructed image.
2. The conformal prediction strategy is employed to calibrate the masking model to obtain strong statistical guarantees.
3. The paper is good writing and easy to follow.

Weakness:
1. The generation way of the continuous mask m is not clear, is it the same as the prevalent implicit neural representation? And the ablation study compared to the binary mask should be provided.
2. The visualization results for super-resolution are not included in the paper.
3. It seems that Figure 2 is dispensable, as it doesn’t provide more meaningful information to understand the experiment setups.


**Summary Of The Paper:**

This paper proposed a new approach for uncertainty estimation in image reconstruction, which uses masking mechanism to identify the more certain regions of the reconstructed image, thus narrowing the distance between masked ground truth and reconstructed images. The experiments are conducted on three reconstruction tasks: image colorization, image completion and super-resolution.

**Summary Of The Review:**

The paper studied the problem of uncertainty estimation in image-to-image domain. The idea seems somewhat interesting, but the effectiveness is not well verified. I think this paper meets the margins of acceptance.

---

> ### Author Response · Authors · 2022-11-17
> **Response to reviewer pBvq**
>
> We thank the reviewer for their comments and suggestions.
>
> **The generation way of the continuous mask m is not clear, is it the same as the prevalent implicit neural representation? And the ablation study compared to the binary mask should be provided.**
>
> We apologize for the lack of clarity; we improved the explanations accordingly. More specifically, the proposed approach is very different from implicit neural representation. Our mask is the regressed output of a trained neural network that gets the degraded image and the recovered one as inputs. The proposed mask assumes continuous values within the range [0,1], conveying the local level of uncertainty. As such, while we could binarize this mask, it will lose critical information and lead to deteriorated uncertainty evaluations. We introduced modifications to the introduction to accommodate for this lack of clarity.
>
> **The visualization results for super-resolution are not included in the paper.**
>
> Thank you for this comment. Our original super-resolution (SR) experiments focused on the Rat Astrocyte Cells dataset, for which we got very good results. However, the involved images lack interpretability and thus we avoided showing visual results of the images and their masks. Following this comment, we added another SR experiment on the XYZ dataset; in the revised version we report the new results and show their outcome visually.
>
> **It seems that Figure 2 is dispensable, as it doesn’t provide more meaningful information to understand the experiment setups.**
>
> We moved the figure to the appendix.

---

### Author Response · Authors · 2022-11-17
**Major modification in the rebuttal version**

Here is a list of the major modifications we did for the original version:
1. We retrained all the models for ten epochs (instead of two) and updated the results and the visual results accordingly.
2. We experimented with the super resolution task on both datasets (Places365 and Rat Astrocyte Cells) and added the results to Table 1 and Figure 2.
3. We moved (previously) Figure 2 to the appendix.
4. Several important clarifications of the theory / algorithm.

---

### Decision · Program_Chairs · 2023-01-20

**Decision:**

Reject

**Justification For Why Not Higher Score:**

Three out of four reviewers recommend rejection, and I have no justification for going against their recommendation.

**Justification For Why Not Lower Score:**

N/A

**Metareview: Summary, Strengths And Weaknesses:**

Summary: This paper presents a new method for estimating uncertainty in image-to-image problems, by training a second model that predicts the "certain" areas of an existing model.

Strengths: The paper tackles a relevant problem and, being post hoc, has broad applicability.

Weaknesses: The method does not predict uncertainty, as claimed by the authors, but rather errors. Clarity could be improved for the more technical parts.